# Interplay between *Arabidopsis thaliana* Genotype, Plant Growth and Rhizosphere Colonization by Phytobeneficial Phenazine-Producing *Pseudomonas chlororaphis*

**DOI:** 10.3390/microorganisms10030660

**Published:** 2022-03-19

**Authors:** Antoine Zboralski, Hara Saadia, Amy Novinscak, Martin Filion

**Affiliations:** 1Saint-Jean-sur-Richelieu Research and Development Centre, Agriculture and Agri-Food Canada, 430 Gouin Boulevard, Saint-Jean-sur-Richelieu, QC J3B 3E6, Canada; antoine.zboralski@agr.gc.ca; 2Department of Biology, Université de Moncton, 18 avenue Antonine-Maillet, Moncton, NB E1A 3E9, Canada; ehs2246@umoncton.ca; 3Agassiz Research and Development Centre, Agriculture and Agri-Food Canada, 6947 Highway 7, Agassiz, BC V0M 1A2, Canada; amy.novinscak@agr.gc.ca

**Keywords:** *Pseudomonas*, phenazine, PGPR, rhizosphere, colonization, rhizocompetence, *Arabidopsis*, SALK, root architecture, immunity

## Abstract

Rhizosphere colonization by phytobeneficial *Pseudomonas* spp. is pivotal in triggering their positive effects on plant health. Many *Pseudomonas* spp. Determinants, involved in rhizosphere colonization, have already been deciphered. However, few studies have explored the role played by specific plant genes in rhizosphere colonization by these bacteria. Using isogenic *Arabidopsis* *thaliana* mutants, we studied the effect of 20 distinct plant genes on rhizosphere colonization by two phenazine-producing *P*. *chlororaphis* strains of biocontrol interest, differing in their colonization abilities: DTR133, a strong rhizosphere colonizer and ToZa7, which displays lower rhizocompetence. The investigated plant mutations were related to root exudation, immunity, and root system architecture. Mutations in *smb* and *shv3*, both involved in root architecture, were shown to positively affect rhizosphere colonization by ToZa7, but not DTR133. While these strains were not promoting plant growth in wild-type plants, increased plant biomass was measured in inoculated plants lacking *fez*, *wrky70*, *cbp60g*, *pft1* and *rlp30*, genes mostly involved in plant immunity. These results point to an interplay between plant genotype, plant growth and rhizosphere colonization by phytobeneficial *Pseudomonas* spp. Some of the studied genes could become targets for plant breeding programs to improve plant-beneficial *Pseudomonas* rhizocompetence and biocontrol efficiency in the field.

## 1. Introduction

Plants play a major role in underground bacterial composition, especially in the rhizosphere, the soil portion under the influence of roots [1,2]. This soil compartment receives rhizodeposits, i.e., nutrients available to the rhizosphere-inhabiting microorganisms, including rhizobacteria, affecting their ability to survive and thrive. These nutrients sustain densities around 10^9^ bacteria per gram of rhizosphere soil, which is 10 to 1000 higher than in bulk soil [3,4]. Depending on the plant species, up to 30% of the carbon fixed by the plant through photosynthesis is released into the rhizosphere [5].

Rhizodeposition relies on numerous factors, such as soil type, plant growth, its physiological state, and root system architecture [6,7,8,9,10]. Rhizodeposits include lysates, volatile compounds, sloughed-off root cells and tissues, as well as root exudates [11]. The latter are mainly released by young tissues surrounding root tips and root hairs, while other rhizodeposits, such as lysates, can be found at sites of lateral root emergence [7,12,13,14]. Root hairs are especially interesting because of their interfacing function between the plant and the rhizosphere: they account for about 77% of the total root surface of crop species and play numerous crucial roles for the plant, such as anchorage, water and nutrient uptake, and exudation, which, in turn, affect the rhizosphere microbiome [7,15,16].

Root exudates are particularly studied because of their diversity and their dynamic release patterns [14]. Different types of exudates are released into the rhizosphere, such as organic acids, sugars, amino acids, or phenolics, influencing the bacterial composition of this habitat [6,17,18]. Exudation is mainly mediated by exocytosis for high-molecular-weight compounds and by passive diffusion through the plasma membrane or active secretion by membrane transporters for lighter metabolites [19]. Different transporters involved in the latter have been identified, belonging to the Major Facilitator Superfamily (MFS), the Multidrug and Toxic Compound Extrusion (MATE) family, the Aluminium-Activated Malate Transporters (ALMT) family, and the ATP-binding cassette (ABC) families [13,20]. Mutations in specific ABC transporters in *Arabidopsis thaliana* have been shown to decrease organic acids exudation or increase amino acids exudation [21]. In tomato, the knockdown of ABC transporters altered the composition of root exudates, which reduced the attraction of a *Bacillus* strain towards exudates [22].

Plant defense signalling can also impact root exudation and bacterial composition in the rhizosphere. Exogenous application of plant immunity signalling molecules on *A. thaliana*, such as salicylic acid, jasmonic acid, and nitric oxide, result in an increase in exudation [23]. It also affects the expression profile of transporters belonging to the MATE and ABC families, as well as to the MFS, highlighting the influence of immunity on root exudation. Other studies focusing on the effect of plant defense signalling mutations affecting the rhizosphere and root bacterial community showed that an altered plant immune system could also affect bacterial colonization [24,25,26].

Rhizosphere colonization is crucial for numerous groups of microorganisms, including plant growth-promoting rhizobacteria (PGPR), involved in plant growth promotion and/or biocontrol of plant pathogens [27,28,29]. Their ability to competitively colonize the rhizosphere and persist in this dynamic environment is defined as rhizocompetence [19,30]. PGPR are able to improve plant nutrition and plant immunity, while producing an array of antimicrobial compounds, directly hampering the growth of plant pathogens and disease development [4]. Among PGPR of interest, *Pseudomonas* spp. have been extensively studied, and a high number of species and strains, displaying strong biocontrol activity, have been described [28,31,32]. *Pseudomonas* spp. are ubiquitous rod-shaped motile Gram-negative bacteria [33], which are found in multiple habitats, thanks to their metabolic versatility [34]. Many of these bacteria can produce antibiotics, including 2,4-diacetylphloroglucinol, pyrrolnitrin, pyoluteorin, and phenazines [29]. Phenazines are heterocyclic and redox-active molecules, capable of efficiently inhibiting a large spectrum of plant pathogens, including several bacteria, fungi and oomycetes [35,36,37]. Some phytobeneficial *Pseudomonas* spp. can produce phenazine-1-carboxylic acid (PCA), phenazine-1-carboxamide (PCN) or 2-hydroxyphenazine (2-OH-PHZ), which have been shown to be involved in the biocontrol of important plant pathogens [38,39]. Phenazines are also involved in biofilm development [40,41] and iron reduction [42,43], facilitating bacterial survival in the rhizosphere [44].

Many rhizocompetence determinants have already been discovered in phytobeneficial *Pseudomonas* spp. [45,46,47], and some studies have highlighted the role played by plant genotypes in rhizosphere colonization [48,49,50]. However, little is known about the role played by specific plant genes in the rhizocompetence of *Pseudomonas* spp. [51,52]. To develop efficient biocontrol strategies, involving promising *Pseudomonas* strains, we need to better understand how the plant genotype can affect the recruitment and the rhizocompetence of phytobeneficial *Pseudomonas* spp. This could lead to new plant breeding targets, enabling phytobeneficial *Pseudomonas* spp. to better colonize the rhizosphere, which, in turn, may benefit plant growth, yield, and protection against pathogens.

The objective of this study was to assess the impact of specific mutations in *A. thaliana* on the rhizosphere colonization capabilities of two distinct phenazine-producing *P*. *chlororaphis* subsp. *piscium* strains of biocontrol interest: DTR133, a strong rhizosphere colonizer, and ToZa7, which displays a lower rhizocompetence [53]. These strains, both isolated from the tomato rhizosphere, were chosen because of their promising traits for biocontrol against different pathogens [54,55,56,57,58], and their distinct rhizosphere colonization levels on *A. thaliana* ecotype Columbia. The studied plant mutants were affected in root exudation, immunity, and root system architecture. The impact of *P. chlororaphis* subsp. *piscium* strains DTR133 and ToZa7 inoculation on plant biomass was also investigated in the selected plant mutants, to better understand the interplay between rhizosphere colonization and plant physiology. This work points to the involvement of several plant genes in rhizosphere colonization and in plant–bacteria interactions.

## 2. Materials and Methods

### 2.1. Plant Cultivation

Seeds of *A. thaliana* ecotype Columbia and 20 isogenic mutants were obtained from the Arabidopsis Biological Resource Center (Columbus, OH, USA) (Table 1). The seeds were sterilized in a bleach and Tween 20 solution for 2 min, then transferred into ethanol 95% for another 2 min, and washed four times in sterile distilled water. They were placed in a 0.1% agar solution and kept at 4 °C for 4 d to enhance and synchronize germination. Sowing was performed in peat-based substrate (Pro-Mix, Premier tech, Rivière-du-Loup, Canada) and in washed and sterilized sand (All-purpose sand, Quickrete, Atlanta, GA, USA) in 10 cm diameter plastic pots. Plants grown in peat-based substrate were used to assess rhizosphere colonization by two *Pseudomonas* strains. Uninoculated sand-grown plants were also used in parallel to phenotypically characterize the impact of the mutations under study. Growth chambers (PGR15, Conviron, Winnipeg, Canada) were used to grow all plants under 60% relative humidity and a 16:8 photoperiod (day:night) at 21 °C, 200 µmol m^−2^ s^−1^, and 20 °C respectively, in a randomized configuration. Plants grown in peat-based substrate were fertilized once, 2 w after sowing, with a 1 g·L^−1^ solution of 20-20-20 fertilizer (All Purpose Fertilizer Water Soluble 20-20-20, Plant-Prod, Brampton, Canada), while those grown in sand received 10% of the same fertilization dose every two days.

### 2.2. Mutant Screening

The *A*. *thaliana* isogenic mutants where chosen based on their phenotypic profiles in relation to root architecture, immunity, or exudation. All investigated mutants were originally generated by T-DNA insertion from *A*. *thaliana* ecotype Columbia [63]. To confirm the presence of the insertion, one leaf was sampled on each plant one day before inoculation of plants grown in peat-based substrate. Plant DNA was extracted following Springer’s protocol [76]. For each DNA sample, two PCR reactions were performed: one targeting the inserted T-DNA, and the other targeting the native gene. Each PCR mixture contained 1 µL of template DNA and was prepared with 2.5 µL of ThermoPol reaction buffer 10× (New England Biolabs, Ipswich, MA, USA), 3 µL of reverse and forward primers and probe (final concentration at 0.6 µmol.L^−1^), and 0.625 units of Taq DNA polymerase (New England Biolabs). Sterile dH_2_O was added for a total reaction volume of 25 µL. The cycling conditions were 95 °C for 5 min followed by 35 cycles of 95 °C for 30 s, annealing for 30 s (for the annealing temperature specific for each gene, see Table A1) and elongation at 72 °C for 2 min. The reaction ended with a final elongation of 10 min at 72 °C. Annealing temperatures were retrieved from the reagent’s manufacturer calculator (https://tmcalculator.neb.com/#!/main, accessed on 23 July 2018). Amplified fragments were verified by electrophoresis on a 1.5% agarose gel. This allowed us to discriminate potential wild-type plants from mutants and heterozygous from homozygous plants. Primer sequences (Table A1) were retrieved from the T-DNA Express website (http://signal.salk.edu/cgi-bin/tdnaexpress, accessed on 13 March 2018) through the iSect tool [77], except for the *pgp1*, the *wrky70* and the *cbp60g* primers, which were designed using Geneious Prime v.2019.2.1 (Biomatters Ltd., Auckland, New Zealand). LBb1.3 was used as left border primer to detect the inserted T-DNA. The primers were custom synthesized by Integrated DNA Technologies (Coralville, IA, USA). Only homozygous plants for a given mutation were further considered.

### 2.3. Bacterial Inoculation in the Rhizosphere

Two strains originally isolated from the tomato rhizosphere were used in this study: *P*. *chlororaphis* subsp. *piscium* DTR133 (a strong rhizosphere colonizer on *A*. *thaliana*) and ToZa7 (a weaker rhizosphere colonizer) [57,58]. They were grown in King’s B broth medium [78] at 120 RPM and 25 °C for 24 h from cryopreserved cultures. Bacterial concentration was assessed with a spectrophotometer at 600 nm (Novaspec II, Pharmacia Biotech, Piscataway, NJ, USA) and adjusted to 2 × 10^8^ CFU ml^−1^. Inoculation was performed on the 20 isogenic mutants and on the wild type. Three-week-old *A*. *thaliana* plants grown in peat-based substrate were inoculated by pipetting 5 mL of the adjusted bacterial suspensions at the base of the stems. Each plant was inoculated with one strain only (or water for the control plants). For each bacterial strain and plant genotype, five replicates were generated for a total of 110 experimental units. This experiment was replicated a second time. Plants grown in sand were not inoculated, as they were used to phenotypically characterize the impact of the mutations under study.

### 2.4. Rhizosphere Harvest and Sample Processing

Three weeks following inoculation of *A*. *thaliana* grown in peat-based substrate, rhizosphere soil was manually harvested by recovering about 30 mL of soil surrounding the roots. The harvested soil was promptly frozen in liquid nitrogen, before being lyophilized (ModulyoD-115, Thermo Electron Corporation, Waltham, MA, USA) and subsequently stored at −80 °C. DNA was then extracted according to Griffiths and colleagues [79], followed by a DNA purification step to remove potential PCR inhibitors (DNeasy PowerClean Cleanup Kit, Qiagen, Hilden, Germany). Aboveground plant parts were also harvested, dried at 70 °C for 10 d and weighed. Roots of plants grown in peat-based substrate could not be harvested and weighed given their thinness, density, and strong adhesion to the substrate.

### 2.5. Bacterial Quantification by qPCR

To estimate *P*. *chlororaphis* subsp. *piscium* DTR133 and ToZa7 rhizosphere colonization, the copy number of the *phzD* gene was assessed by quantitative PCR (qPCR) from all rhizosphere soil DNA samples. Encoding an isochorismatase, this gene belongs to the phenazine biosynthetic cluster [37]. A TaqMan probe and a primer pair designed in a previous study targeting both *Pseudomonas* strains under study were used, specifically generating a 83 bp-amplicon [53]. The TaqMan probe was labeled with a 6-carboxyfluorescein (6-FAM) reporter dye at the 5’ end and a non-fluorescent minor groove-binding quencher at the 3’ end (MGBNFQ, Applied Biosystems, Waltham, MA, USA). The primers were custom synthesized by Integrated DNA Technologies. The sequences used to amplify the *phzD* 83 bp-amplicon are: probe, GAA TAC GCC GCC AGC; forward primer, GCA AGG MGC AYC ACT GGA T; reverse primer, TCA TAG CAB CAC CTC RTC GG.

To standardize quantification, the 83-bp *phzD* PCR amplicon was inserted into the 2,976-bp pKRX plasmid (National Institute of Genetics, Mishima, Japan). The resulting plasmid was transferred into *E*. *coli* to be replicated before being extracted using a kit (QIAprep Spin Miniprep, Qiagen) and checked on agarose gel for an insert of the correct size following PCR amplification. The amount of plasmid DNA was assessed using a fluorometer (Qubit 3 Fluorometer, Thermo Fisher Scientific, Waltham, MA, USA). The number of plasmid copies was then inferred by considering the molar mass of the modified pKRX plasmid containing the *phzD* amplicon. Amplicon-containing plasmid DNA was then serially diluted to known concentrations to generate standard curves ranging from 10^8^ to 10^1^ gene copies µL^−1^.

A qPCR bioassay using the *phzD* gene as a target was performed with a Bio-Rad CFX Connect real-time PCR detection system and the iTaq universal probe supermix kit (Bio-Rad Laboratories, Hercules, CA, USA). Each qPCR mixture was prepared with 5 µL (1×) of iTaq universal probe supermix, 0.4 µL of reverse and forward primers and probe (final concentration at 0.2 µmol.L^−1^), 2.8 µL of sterile dH_2_O, and 2.4 µL of tenfold-diluted template DNA, for a total volume of 10 µL. The cycling conditions were 95 °C for 2 min followed by 40 two-step cycles consisting of 95 °C for 5 s and 60 °C for 30 s. No template controls were included in each qPCR run by adding sterile dH_2_O instead of DNA. All qPCRs were replicated 3 times. Standard curves were generated for each 96-well qPCR plate. *phzD* copy numbers were adjusted per gram of rhizosphere soil. The primers and the probe used to quantify the bacterial strains were tested against uninoculated soil and showed no amplification.

### 2.6. Phenotypic Characterization of the Root System of Sand-Grown Plants

The 20 isogenic plant mutants used in this study and the wild type were grown in sand, as described above, to facilitate the retrieval of the entire root system for phenotypic characterization. The plants were harvested 6 weeks after sowing, the same period as the plants grown in the peat-based substrate that were inoculated with bacteria. The root system was washed in water to separate sand particles from it. It was then put in a transparent tank filled with water, and photographed (Lumix DMC-ZX1, Panasonic, Kadoma, Osaka, Japan). Some roots were cut, and representative root hair zones were photographed using a microscope (Leitz DMRB and MC170 HD, Leica Microsystems GmbH, Wetzlar, Germany). The roots and aboveground parts were dried at 70 °C for 10 d and weighed. For each plant genotype, 3 replicates were used for a total of 63 experimental units.

### 2.7. Statistical Analyses

All statistical analyses were performed using RStudio version 1.2.5001 (Boston, MA, USA). The ‘agricolae’ package version 1.3-1 [80] and the ‘nparcomp’ package version 3.0 [81,82] were used to perform multiple comparisons using a Benjamini–Hochberg correction. Unless stated otherwise, test results were interpreted at a 5% significance level. Inter-experiment effects were considered.

## 3. Results

### 3.1. Rhizosphere Colonization of Mutant Plants by Both Pseudomonas Strains

To assess rhizosphere colonization by the two *P. chlororaphis* subsp. *piscium* strains, DTR133 and ToZa7, twenty distinct *A. thaliana* isogenic mutants (and the wild type) affected in root exudation, immunity, and root system architecture were grown in non-sterile potting soil and inoculated. A culture-independent qPCR approach, targeting a conserved DNA motif in the *phzD* gene, was used to specifically detect and quantify *P. chlororaphis* subsp. *piscium* strains DTR133 and ToZa7 in the rhizosphere, three weeks following inoculation (Figure 1). Uninoculated soil was also tested to ensure the specificity of the qPCR bioassay, which led to no amplification. Mutations in plants were all confirmed using PCR with primers specific to the inserted T-DNA.

ToZa7 colonized the rhizosphere of plants impaired in *smb* and *shv3* on average four-times more than wild-type plants (Figure 1A). When plants were inoculated with DTR133, rhizosphere colonization did not statistically differ between mutants and wild-type plants (Figure 1B). As expected, colonization of wild-type *A. thaliana* was significantly higher for DTR133 than for ToZa7, on average with a 16-times difference (Figure 1C). The colonization difference between both strains was significant in 19 mutant genotypes (Figure 1C). In those genotypes, DTR133 colonized the rhizosphere 4- to 32-times more than ToZa7. In *lox1* mutant plants only, no significant difference in rhizosphere colonization pattern was observed between DTR133 and ToZa7 (Figure 1C). This loss in colonization advantage for DTR133 seems to arise from its reduced colonization capabilities, rather than to an increased colonization by ToZa7 (Figure 1B).

### 3.2. Aboveground Biomass Accumulation in the Inoculated Mutants Plants

Three weeks following inoculation, plant aboveground parts were harvested, dried, and weighed (Figure 2). Wild-type plants, inoculated with either bacterial strain, did not significantly differ in aboveground biomass, indicating that neither *P. chlororaphis* subsp. *piscium* strain DTR133, nor ToZa7 can be considered plant growth promoters on wild-type *A. thaliana* (Figure 2A). Some mutations, however, had a negative impact on the biomass of uninoculated plants (Figure 2B). Without bacterial inoculation, seven plant genotypes (*xik*, *rlp30*, *teb*, *wrky70*, *shv3*, *arf7*, and *pft1*) exhibited an 11% to 33% significant mass reduction, when compared to the wild type.

Significant plant biomass differences were found between inoculation treatments for each plant mutant genotype. Seven genotypes displayed increased aboveground biomass when inoculated with DTR133 or ToZa7, compared to water-treated plants (Figure 2A). This increase ranged from 13% to 31% and relates to *rlp30*, *cbp60g*, *shv3*, *smb*, *fez*, *pft1*, and *wrky70* plants. Two genotypes instead displayed decreased plant biomass when inoculated with DTR133 or ToZa7: *pgp1* and *tor*. This decrease ranged from 15% to 25%. For most of these mutants displaying different biomasses when inoculated, inoculation had a similar effect in comparison to wild-type plants, whether they were inoculated by DTR133 or ToZa7. However, two plant mutants reacted differently: *tor* and *rlp30*. These plants displayed a biomass distinct from wild-type plants when inoculated with one strain, but not with the other. All these results point to a role for these genes in plant–bacteria interactions.

### 3.3. Phenotypic Characterization of the Mutants Grown in Sterile Sand

To better assess how mutations could impact on the root system and the rhizosphere, the 21 plant genotypes were grown in sterile sand, without any bacterial inoculation. This allowed for a clear separation between the root system and the substrate, where the mass of the roots and of the aboveground parts can be separately measured (Figure 3), and microscopic observations of the root system can be achieved (Figure 4). Only one mutation significantly impacted the root dry mass compared to the wild type: *shv3* (Figure 3A). On average, the dried root system of *shv3* plants was about five-times lighter than the wild type. When examined under the microscope, *shv3* roots exhibited shorter root hairs than the wild type (Figure 4), which was not the case for the other mutants under study.

For the aboveground biomass part of plants grown in sterile sand, only a single genotype displayed a significant difference with the wild type: *teb*. Plants impaired in *teb* displayed, on average, 45% higher aboveground biomass parts than the wild type. When instead grown in peat-based substrate without inoculation, plants impaired in *teb* displayed on average 14% lower biomass than the wild type (Figure 3B).

## 4. Discussion

This study aimed at evaluating the effect of specific mutations in *A. thaliana* on rhizosphere colonization, by two phenazine-producing *P*. *chlororaphis* subsp. *piscium* strains of biocontrol interest, DTR133 and ToZa7, known for their distinct colonization abilities on *A. thaliana*. The rhizosphere colonization of 20 different *A. thaliana* isogenic mutants (and the wild type) was assessed in non-sterile potting soil, three weeks following bacterial inoculation. In parallel, plant aboveground biomass was measured to further assess the interactions existing between bacterial inoculation treatments and specific *A. thaliana* mutations. The *A. thaliana* isogenic mutants where chosen based on their phenotypic profiles, in relation to root architecture, immunity, or exudation.

### 4.1. Rhizosphere Colonization Is Affected by the Plant Genotype

Plants impaired in *smb* and *shv3* displayed higher rhizosphere colonization by ToZa7 than wild-type plants (Figure 1A). SOMBRERO (SMB) is a NAC-domain (for NAM, ATAF1/2, and CUC2) transcription factor controlling the root cap development [61,69]. When plants are impaired in *smb*, lateral root cap maturation is delayed, and the root cap extends into the differentiation zone instead of sloughing off the root tip. Root caps produce detached active cells, called border cells, involved in exudation and plant defense [83]. Mutations in *smb* may affect these cells, especially their exudation profiles or their anti-microbial role, leading to increased bacterial colonization by ToZa7. SHAVEN3 (SHV3) is a glycerophosphoryl diester phosphodiesterase-like protein, involved in cell wall organization and root hair morphogenesis [15,68]. When *shv3* is impaired, root hair tip growth is blocked because of ruptures in root hair cells, as illustrated by our microscopic observations (Figure 4). The presence of shorter root hairs may explain the lighter root system observed in *shv3* plants, in comparison to the wild type (Figure 3A). Root hairs represent up to 77% of root surface in cultivated crops and are, with the root caps, important sources of root exudates [16,84]. An altered root system architecture may improve plant–bacteria interactions, leading to an increased rhizosphere colonization. Contrary to ToZa7, DTR133 colonized the rhizosphere of mutant plants to the same extent as the wild-type plants (Figure 1B). It may, therefore, display a different colonization strategy than ToZa7, thus, differently interacting with the mutant phenotypes.

As previously shown, there is a significant rhizosphere colonization-level difference between both bacterial strains in *A. thaliana* wild-type plants [53]. *P. chlororaphis* subsp. *piscium* strain DTR133 is a strong rhizosphere colonizer, while ToZa7 displays a lower rhizocompetence. This significant difference in rhizosphere colonization patterns was also observed in 19 out of the 20 *A. thaliana* mutants under study. Interestingly, no colonization differences between the two *P. chlororaphis* strains were observed in one *A. thaliana* mutant: *lox1* (Figure 1C). LOX1 is a lipoxygenase, initiating the biosynthesis of plant oxylipins, which are lipid derivatives, such as jasmonic acid, involved in plant physiological processes, including growth and fertility [65]. It may also be involved in stress signalling from roots to shoots [85]. Plants impaired in LOX1 develop more emergent and lateral roots than the wild type. In *lox1* mutants, the absence of colonization differences between both *P. chlororaphis* strains seems to arise from a reduced colonization by DTR133, rather than an increased colonization by ToZa7 (Figure 1B). This differential impact of *lox1* mutant plants on rhizosphere colonization by *P. chlororaphis* strains could be explained by distinct colonization patterns interacting with the altered root system architecture. Noirot-Gros et al. have shown that two *P. fluorescens* strains, SBW25 and WH6, had distinct patterns of biofilm formation on Aspen roots [86]. Further, Pliego et al. demonstrated that *P. alcaligenes* AVO73 and *P. pseudoalcaligenes* AVO110 had distinct avocado root colonization strategies [87]. It would be interesting, in a follow-up study, to assess ToZa7 and DTR133 colonization strategies in the rhizosphere of *lox1* mutant plants, compared to the wild type, for example, by using fluorescent labeling and confocal microscopy. Interestingly, LOX1 is involved in jasmonic acid-dependant plant defense pathways. Jasmonic acid and its derivatives are known for mediating beneficial plant–bacteria interactions, especially through induced systemic resistance (ISR) [88]. ToZa7 has been shown to induce the expression of defense-related genes, potentially involved in ISR [56]. To our knowledge, DTR133 has not been shown to induce such a mechanism yet. Jasmonic acid could play a critical role in its interaction with *A. thaliana*, affecting rhizosphere colonization.

### 4.2. Bacterial Inoculation on Several Plant Mutants Differently Impacts Plant Biomass

Plants impaired in *cbp60g*, *shv3, smb*, *fez*, *pft1*, or *wrky70*, inoculated with ToZa7 or DTR133, displayed increased biomass compared to water-treated plants (Figure 2A). These genes are mostly involved in the plant immunity. ToZa7, the less efficient rhizosphere colonizer, was shown to better colonize the rhizosphere of *smb* and *shv3* plants than the wild type, which could potentially improve its interactions with *A. thaliana* and increase biomass. This was, however, not the case for DTR133, for which no significant colonization differences between the 20 mutants under study were observed. ToZa7 and DTR133 have previously been shown to harbor genes related to plant growth promotion, involved in the biosynthesis of auxin, pyrroloquinoline quinone, and 2,3-butanediol, that could potentially explain the biomass increase observed in different *A. thaliana* mutants [89]. However, the mechanisms underlying plant growth promotion in mutants and not in the wild type remain to be elucidated. FEZ, like SMB, is a NAC-domain transcription factor, controlling the root cap development [61,69]. FEZ and SMB regulate each other, controlling stem cells divisions in the root cap. Plants impaired in *fez* display fewer columella and lateral root cap cell layers. The effect of both mutations on root structure may have affected the *Pseudomonas* spp. colonization process and their ability to impact the plant by modifying root tip exudation or root colonization sites. WRKY70 is a transcription factor, playing a crucial role in the balance between salicylic acid- and jasmonic acid-dependant defense pathways in *A. thaliana* [73]. Plants impaired in *wrky70* are more susceptible to bacterial, fungal, and oomycete pathogens [90,91]. Further, *wrky70* has been shown to be upregulated, following rhizosphere inoculation with PGPR, such as *P*. *fluorescens* SS101 [92] and *Bacillus cereus* AR156 [93], leading to induced systemic resistance against *P*. *syringae* pv. *tomato*. The disruption of *wrky70* could impair the *A. thaliana* immune system and enhance plant–bacteria interactions, leading to an increased biomass. CBP60g (calmodulin-binding protein 60-like g) is a transcription factor involved in MAMP-triggered (microbe-associated molecular patterns) immunity, especially in salicylic acid signalling [60]. It has been proposed to function as a master regulator of the plant immune system [94]. Plants impaired in *cbp60g* display enhanced growth of the bacterial phytopathogen *P. syringae*. PFT1 (phytochrome and flowering time, also called MED25) is a subunit of the Mediator complex, a large multiprotein complex involved in transcription regulation [66]. PFT1 plays a role in jasmonate-dependent plant defenses, but is also involved in root hair development [95]. Considering these results, we hypothesise that an altered plant immune response, due to at least some of these mutations, may improve plant–bacteria interactions, leading to an increased plant biomass.

Plants impaired in *rlp30*, inoculated with DTR133, displayed increased biomass compared to water-treated plants, while inoculation with ToZa7 had no significant effect on biomass (Figure 2A). RLP30 is a receptor-like protein, located at the cell surface, involved in disease resistance [67]. Mutant plants display enhanced susceptibility against *P. syringae* pv *phaseolicola*, *Sclerotinia sclerotiorum*, and *Botrytis cinerea*, probably because of a defect in the plant MAMP-triggered immune response [96]. This mutation seems to specifically affect the interaction between DTR133 and the plant. DTR133 may harbor an elicitor of the plant immune system that ToZa7 does not produce. It may, thus, not be recognized by the mutant plant anymore, allowing the bacteria to better interact with it in the rhizosphere.

Bacterial inoculation of plants impaired in *pgp1* or *tor* negatively impacted plant biomass. PGP1 (multi-drug resistance P-glycoprotein) is an ATP-binding cassette transporter involved in auxin transport [21]. Plants impaired in *pgp1* display more lateral roots, as well as an altered root exudation profile. TOR (target of rapamycin) is a conserved eukaryotic kinase regulating cell growth, according to nutrient availability [71]. Plants harboring a T-DNA upstream *TOR* (accession SALK_007846) overexpress *TOR* mRNA in roots, leading to increased root growth. Deprost et al. suggested that this is probably caused by a strong promoter, carried by the inserted T-DNA [71]. Bacterial inoculation might lead to an imbalance between root growth and shoot growth, in favor of the root system, which could be detrimental to the accumulation of biomass in the aboveground plant parts.

### 4.3. Changes in Root Exudates Composition Do Not Alter Rhizosphere Colonization

Badri et al. showed that *pgp1* and *mrp2* mutant plants display altered root exudates compositions [21], which could have impacted rhizosphere colonization levels of DTR133 and ToZa7 or their colonization trends. Results obtained in this study did not support this hypothesis: both strains inoculated in the rhizosphere of these mutants showed no significant difference in colonization compared to the wild type. However, inoculated *pgp1* plants displayed reduced biomass compared to water-treated plants. Because rhizosphere colonization was not significantly impacted by this mutation, this decrease probably arose from changes in plant–bacteria interactions.

### 4.4. The teb Mutation Interplays with the Plant Substrate to Affect Aboveground Biomass Accumulation

Uninoculated plants, impaired in *teb*, demonstrated a higher aboveground biomass accumulation than the wild type when grown in sand (Figure 3B), while they showed reduced biomass accumulation compared to the wild type when grown in the peat-based substrate (Figure 2B). TEBICHI is a DNA polymerase, involved in the plant DNA damage responses [97]. Plants lacking this protein display shorter roots and a smaller aerial system [70]. The differential impact of substrates on biomass accumulation in this mutant remains unexplained.

The various *A. thaliana* mutants used in this study have been generated by different research teams, who phenotypically characterized them according to their own topics of interest. The information available for these plants is consequently heterogeneous. Further phenotypic characterization of these mutants could strengthen our understanding of the underlying plant–bacteria interaction mechanisms, especially regarding plant growth promotion and biocontrol.

In conclusion, this study aimed at evaluating the impact of specific mutations in *A. thaliana* on rhizosphere colonization by two distinct phenazine-producing *P. chlororaphis* subsp. *piscium* strains of biocontrol interest, and their interacting impacts on plant biomass. Mutations in two plant genes, involved in the root system architecture, *smb* and *shv3*, have been shown to positively impact ToZa7 rhizosphere colonization and to allow for biomass increase, when bacterial inoculation was performed. The inactivation of another gene, related to the root system architecture, *lox1*, induced a change in colonization patterns between ToZa7 and DTR133. Mutations in several genes, mostly involved in the plant immune system, were associated with biomass increase only, in relation to bacterial inoculation: *fez*, *wrky70*, *cbp60g*, *pft1*, and *rlp30*. The results indicate an interplay between plant genotype, plant growth and bacterial rhizosphere colonization. Further investigation will be required to decipher the mechanisms underlying the impact of these genes on rhizospheric bacteria colonization, to improve *Pseudomonas* spp. rhizocompetence, and ultimately, their biocontrol abilities in the field. Some of the candidate genes studied here could become interesting targets for assisted plant breeding programs.

## Figures and Tables

**Figure 1 microorganisms-10-00660-f001:**
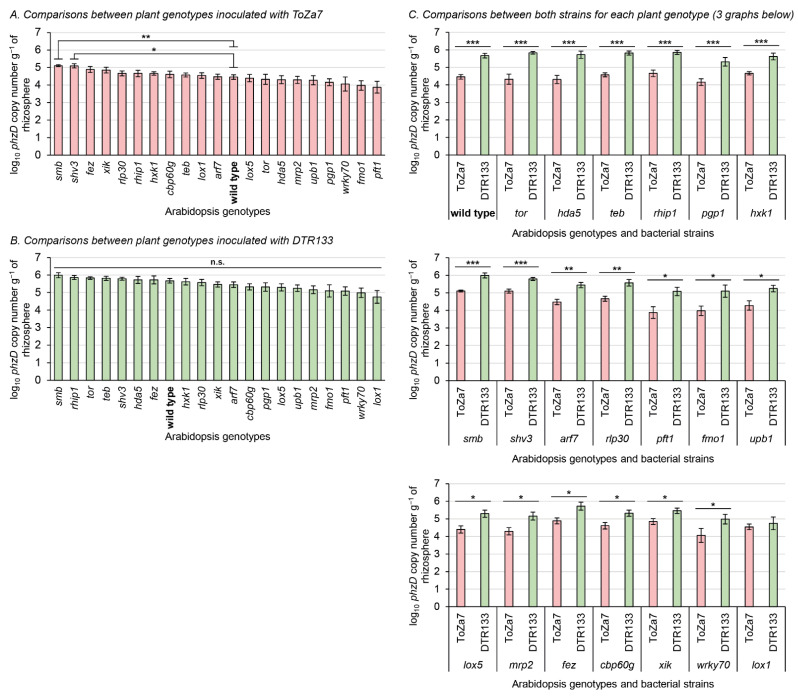
Rhizosphere colonization three weeks following bacterial inoculation with *P*. *chlororaphis* subsp. *piscium* strains ToZa7 and DTR133 for 21 *A*. *thaliana* genotypes. Asterisks refer to significant differences between groups. (**A**,**B**): nonparametric multiple test procedure for many-to-one comparisons [81], allowing the comparison of all mutants against the wild type. (**C**): Wilcoxon–Mann–Whitney tests. * *p*-value < 0.05; ** *p*-value < 0.01; *** *p*-value < 0.001; n.s.: non-significant. Bars indicate standard errors (*n* = 10).

**Figure 2 microorganisms-10-00660-f002:**
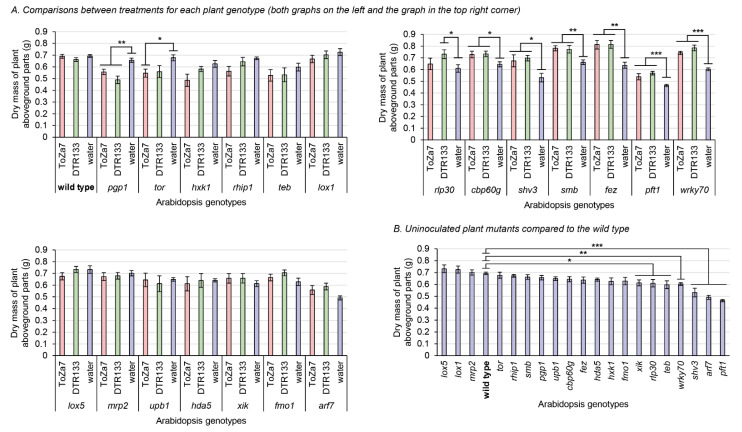
Dry mass of plant aboveground parts three weeks following inoculation with *P*. *chlororaphis* subsp. *piscium* strains ToZa7 and DTR133 for 21 *A*. *thaliana* genotypes. Asterisks refer to significant differences between groups defined by Fisher’s least significant difference test. * *p*-value < 0.05; ** *p*-value < 0.01; *** *p*-value < 0.001. Bars indicate standard errors (*n* = 10). (**A**) Comparisons between treatments for each plant genotype (graphs have been divided for easy layout). (**B**) Uninoculated plant mutants compared to the wild type.

**Figure 3 microorganisms-10-00660-f003:**
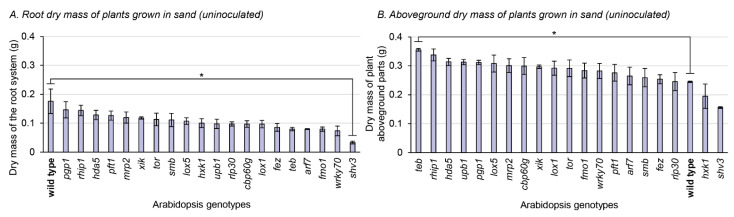
Dry mass of roots and aboveground parts of 21 *A*. *thaliana* genotypes grown in sterile sand (not inoculated). Asterisks refer to significant differences between the wild type and another genotype, defined by Fisher’s least significant difference test. * *p*-value < 0.05. Bars indicate standard errors (*n* = 3). (**A**) Root dry mass of plants grown in sand (uninoculated). (**B**) Aboveground dry mass of plants grown in sand (uninoculated).

**Figure 4 microorganisms-10-00660-f004:**
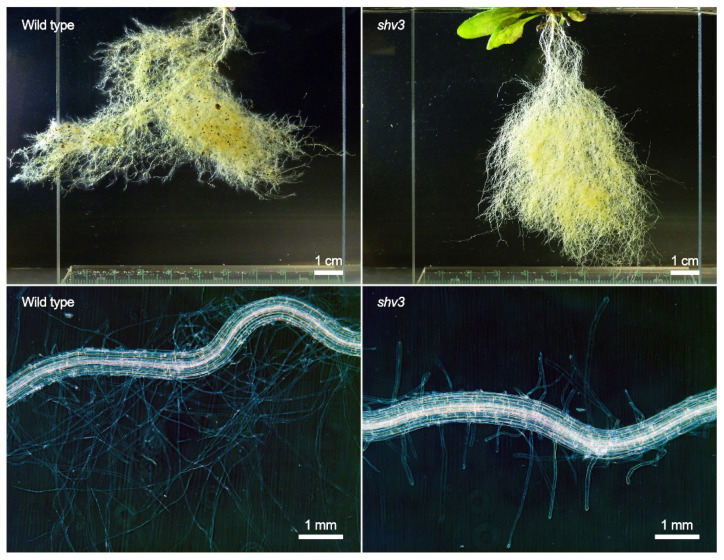
Root system (**top**) and root hair zone (**bottom**) of representative uninoculated *A*. *thaliana* wild type (**left**) and *shv3* mutant plants (**right**) grown in sterile sand.

**Table 1 microorganisms-10-00660-t001:** Genotypes of *A. thaliana* used in this study. All mutant strains derive from Col-0. Additional characteristics are available on The Arabidopsis Information Resource website (https://www.arabidopsis.org/, accessed on 13 March 2018).

Affected Gene	Related Protein	Phenotypic Impacts of the Mutation	Accession Number	Reference
wild type	-	-	CS70000 (Col-0)	-
*arf7*	AUXIN RESPONSE FACTOR 7	Impaired phototropic and gravitropic response in hypocotyls, reduced numbers of lateral roots, epinastic rosette leaves, reduced auxin sensitivity in hypocotyl growth	CS24607	[59]
*cbp60g*	CAM-BINDING PROTEIN 60-LIKE G	Enhanced susceptibility to *P. syringae*	SALK_023199C	[60]
*fez*	FEZ	Reduced number of cell layers in the root caps	SALK_025663C	[61]
*fmo1*	FLAVIN-DEPENDENT MONOOXYGENASE 1	Increased susceptibility to virulent *P. syringae* infection	SALK_026163C	[62]
*hda5*	HISTONE DEACETYLASE 5	Increased root hair density	SALK_093312C	[63]
*hxk1*	HEXOKINASE 1	Reduced growth in roots, leaf and stem length, rosette size and inflorescence	CS69135	[64]
*lox1*	LIPOXYGENASE 1	Moderate increase in the length of the primary root and increased number of emergent and lateral roots	SALK_059431C	[65]
*lox5*	LIPOXYGENASE 5	Moderate increase in the length of the primary root and increased number of emergent and lateral roots	SALK_044826C	[65]
*mrp2*	MULTIDRUG RESISTANCE- ASSOCIATED PROTEIN 2	Changes in root exudates composition and shorter primary root	CS66052	[21]
*pft1*	PHYTOCHROME AND FLOWERING TIME 1	Enhanced susceptibility to leaf infecting necrotrophic pathogens	SALK_129555C	[66]
*pgp1*	P-GLYCO- PROTEIN 1	Changes in root exudates composition and increased lateral root formation	CS66051	[21]
*rhip1*	RGS1-HXK1 INTERACTING PROTEIN 1	Longer roots in young seedlings and larger inflorescence	CS69137	[64]
*rlp30*	RECEPTOR LIKE PROTEIN 30	Enhanced susceptibility to bacterial and fungal pathogens	CS65465	[67]
*shv3*	SHAVEN 3	No tip growth in almost all root hair cells	SALK_024208C	[68]
*smb*	SOMBRERO	Additional cell layers in the root caps	SALK_143526C	[61,69]
*teb*	TEBICHI	Short-root phenotype and reduced size of the aerial system, with highly serrated and asymmetric leaves and a fasciated stem	SALK_018851C	[70]
*tor*	TARGET OF RAPAMYCIN	Larger plants, more resistant to osmotic stress	SALK_007846C	[71]
*upb1*	UPBEAT 1	Longer roots, increased number of cortex cell	CS868100	[72]
*wrky70*	WRKY DNA-BINDING PROTEIN 70	Enhanced susceptibility to bacterial and fungal pathogens	SALK_025198C	[73]
*xik*	MYOSIN XI-K	Reduced length and altered shape in root hair	SALK_067972C	[74,75]

## Data Availability

Data are contained within the article.

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
