# Peer review of "Interplay between *Arabidopsis thaliana* Genotype, Plant Growth and Rhizosphere Colonization by Phytobeneficial Phenazine-Producing *Pseudomonas chlororaphis"

_microorganisms, 2022, doi:10.3390/microorganisms10030660_

Round 1
Reviewer 1 Report
In the present study, the effect of plant genes on rhizosphere colonization by two phenazine-producing P. chlororaphis strains, potential plant growth stimulation, and biocontrol agent. The research work presents an interesting topic and is well organized. I have provided some comments and suggestions, which may help authors improve their Manuscript.
Comments
Lines 123: “Plants grown in the peat-based substrate were fertilized once with a full fertilizer dose,”, Please add more details about fertilizers used for plant nutrition.
Line 170: “The A. thaliana isogenic mutants” were used for experiments as described in Table 1. Are they used for bacterial inoculation?
Line 177: Plants were grown in the peat-based substrate, and roots could not be harvested, but the 20 isogenic plant mutants were grown in the sand only, are they also inoculated? (line 215). In this experiment, root systems were analyzed. Please clarify this part.
Line 236: Is root exudation were also studied?
The colonization ability of bacteria in the rhizosphere of A. thaliana isogenic mutants is different. Did you also study the exudate composition of mutants, which could affect bacterial colonization? Competition for nutrient and niches among rhizosphere bacteria is high, and it would be interesting to see if exudate composition support their proliferation.
Moreover, the potential colonization ability of bacterial inoculants in the rhizosphere in the non-sterile condition is also important for biological control. It would be good to discuss some related reports on the competitive colonization abilities of inoculants in natural soil. Phenazine producing Pseudomonas chlororaphis were used in this study, while colonising plant root, they produce various metabolites. Is there any effect of phenazine on plant development?
Author Response
In the present study, the effect of plant genes on rhizosphere colonization by two phenazine-producing P. chlororaphis strains, potential plant growth stimulation, and biocontrol agent. The research work presents an interesting topic and is well organized. I have provided some comments and suggestions, which may help authors improve their Manuscript.
Lines 123: “Plants grown in the peat-based substrate were fertilized once with a full fertilizer dose,”, Please add more details about fertilizers used for plant nutrition.
> The sentence was amended accordingly: “Plants grown in peat-based substrate were fertilized once, 2 w after sowing, with a 1 g.l-1 solution of 20-20-20 fertilizer”.
Line 170: “The A. thaliana isogenic mutants” were used for experiments as described in Table 1. Are they used for bacterial inoculation?
> The following sentence was added: “Inoculation was performed on the 20 isogenic mutants and on the wild type.”
Line 177: Plants were grown in the peat-based substrate, and roots could not be harvested, but the 20 isogenic plant mutants were grown in the sand only, are they also inoculated? (line 215). In this experiment, root systems were analyzed. Please clarify this part.
> The following sentence was added at the end of the “Bacterial inoculation in the rhizosphere” section to provide clarification: “Plants grown in sand were not inoculated, as they were used to phenotypically characterize the impact of the mutations under study”.
Line 236: Is root exudation were also studied? The colonization ability of bacteria in the rhizosphere of A. thaliana isogenic mutants is different. Did you also study the exudate composition of mutants, which could affect bacterial colonization? Competition for nutrient and niches among rhizosphere bacteria is high, and it would be interesting to see if exudate composition support their proliferation.
> Collecting and characterizing root exudates would indeed be an interesting approach to better understand the colonization dynamics of the bacteria under study. While our work did not specifically address this aspect, some of the plant mutants used in our study (pgp1 and mrp2) have previously been shown to display altered exudation profiles. The colonization results obtained have been discussed accordingly in the discussion.
Moreover, the potential colonization ability of bacterial inoculants in the rhizosphere in the non-sterile condition is also important for biological control. It would be good to discuss some related reports on the competitive colonization abilities of inoculants in natural soil.
> Absolutely, rhizosphere colonization abilities in non-sterile soils are important for biocontrol. Nonetheless, the focus of this study was on plant genotypes and their effects on bacterial colonization. While competition in the rhizosphere with other microbes certainly occurred under our experimental conditions, we chose to focus on plant-biocontrol bacteria interactions and not on microbe-microbe competition within the peat-based substrate.
Phenazine producing Pseudomonas chlororaphis were used in this study, while colonising plant root, they produce various metabolites. Is there any effect of phenazine on plant development?
> To our knowledge, phenazines (PCA and PCN) produced by the bacterial strains under study do not directly affect plant development. However, they play many roles for the producing bacteria, especially in biofilm formation, antibiosis, and redox balancing, ultimately improving their fitness.
Reviewer 2 Report
I congratulate the authors for their meticulous work.
Manuscript microorganisms-1633273 "Interplay between Arabidopsis thaliana genotype, plant growth and rhizosphere colonization by phytobeneficial phenazine-producing Pseudomonas chlororaphis" by Zboralski and collaborators is very well written and scientifically sound. Appropriate experimental design and statistical analyses were needed to compare so many Arabidopsis genotypes. The Introduction provides enough background and the Discussion highlights possible future directions. The topic of plant growth-promoting bacteria is of interest to many readers of Microorganisms. The manuscript seems ready for publication and I recommend acceptance in the current format.
Author Response
I congratulate the authors for their meticulous work.
Manuscript microorganisms-1633273 "Interplay between Arabidopsis thaliana genotype, plant growth and rhizosphere colonization by phytobeneficial phenazine-producing Pseudomonas chlororaphis" by Zboralski and collaborators is very well written and scientifically sound. Appropriate experimental design and statistical analyses were needed to compare so many Arabidopsis genotypes. The Introduction provides enough background and the Discussion highlights possible future directions. The topic of plant growth-promoting bacteria is of interest to many readers of Microorganisms. The manuscript seems ready for publication and I recommend acceptance in the current format.
> Thank you for your comments.